# Preliminary evaluations of 3-dimensional human skin models for their ability to facilitate *in vitro* the long-term development of the debilitating obligatory human parasite *Onchocerca volvulus*

**Christoph Malkmus**[1]☯, **Shabnam Jawahar**[2]☯, **Nancy Tricoche**[2], **Sara Lustigman**[2]*, **Jan Hansmann**[3]*

**1** Tissue Engineering and Regenerative Medicine, University Hospital Würzburg, Würzburg, Germany, **2** Molecular Parasitology, Lindsley F. Kimball Research Institute, New York Blood Center, New York, New York, United States of America, **3** Translational Center for Regenerative Therapies, Fraunhofer ISC, Würzburg, Germany

☯ These authors contributed equally to this work.
* SLustigman@nybc.org (SL); jan.hansmann@isc.fraunhofer.de (JH)

## Abstract

Onchocerciasis also known as river blindness is a neglected tropical disease and the world's second-leading infectious cause of blindness in humans; it is caused by *Onchocerca volvulus*. Current treatment with ivermectin targets microfilariae and transmission and does not kill the adult parasites, which reside within subcutaneous nodules. To support the development of macrofilaricidal drugs that target the adult worm to further support the elimination of onchocerciasis, an in-depth understanding of *O. volvulus* biology especially the factors that support the longevity of these worms in the human host (>10 years) is required. However, research is hampered by a lack of access to adult worms. *O. volvulus* is an obligatory human parasite and no small animal models that can propagate this parasite were successfully developed. The current optimized 2-dimensional (2-D) *in vitro* culturing method starting with *O. volvulus* infective larvae does not yet support the development of mature adult worms. To overcome these limitations, we have developed and applied 3-dimensional (3-D) culture systems with *O. volvulus* larvae that simulate the human *in vivo* niche using *in vitro* engineered skin and adipose tissue. Our proof of concept studies have shown that an optimized indirect co-culture of *in vitro* skin tissue supported a significant increase in growth of the fourth-stage larvae to the pre-adult stage with a median length of 816–831 μm as compared to 767 μm of 2-D cultured larvae. Notably, when larvae were co-cultured directly with adipose tissue models, a significant improvement for larval motility and thus fitness was observed; 95% compared to 26% in the 2-D system. These promising co-culture concepts are a first step to further optimize the culturing conditions and improve the long-term development of adult worms *in vitro*. Ultimately, it could provide the filarial research community with a valuable source of *O. volvulus* worms at various developmental stages, which may accelerate innovative unsolved biomedical inquiries into the parasite's biology.

**Data Availability Statement:** All relevant data are within the manuscript and its Supporting Information files.

**Funding:** This work was supported by National Institutes of Health (https://www.nih.gov/) grant number 1R21AI131701 to SL. The funders had no role in study design, data collection and analysis, decision to publish, or preparation of the manuscript.

**Competing interests:** The authors have declared that no competing interests exist.

## Author summary

The filarial nematode *Onchocerca volvulus* is an obligatory human parasite and the causative agent of onchocerciasis, better known as river blindness. In 2017, more than 20 million infections with *O. volvulus* were estimated worldwide, 99% of the patients live in Africa. Current international control programs focus on the reduction of microfilaridermia by mass drug administration of ivermectin. However, to meet the elimination goals, additional treatment strategies are needed that also target the adult worms. As this parasite is obliged to humans, there are no small animal models that sustain the full life cycle of the parasite, thus greatly impeding the research on this filarial nematode. To overcome these drawbacks, we have developed co-culture systems based on engineered human skin and adipose tissue that represent the *in vivo* niche of *O. volvulus* adult worms that improved the culturing conditions and the development to the pre-adult stages of the parasite. Furthermore, our new culture approach could significantly reduce the use of surrogate animal models currently used for macrofilaricidal drug testing.

## Introduction

The filarial nematode *Onchocerca volvulus* is an obligatory human parasite and the causative agent of onchocerciasis. This neglected tropical disease also known as river blindness predominantly occurs in sub-Saharan Africa [1,2]. Infectious *O. volvulus* third stage larvae (L3; S1 Text) are transmitted by infected black flies to humans by entering the skin through the wound created during the vector's blood meal. The larvae migrate to subcutaneous tissues, where they further develop to adult worms within subcutaneous highly vascularized nodular tissues consisting of extracellular matrix (ECM) and various immune cells, such as macrophages, neutrophils and eosinophils [3–5]. Inside these nodules, the reproduction of male and female worms results in the release of large numbers of microfilariae that migrate into the surrounding skin. Dead microfilariae and the resulting inflammatory reactions often cause the *Onchocerca* skin disease. When microfilariae reach ocular tissues, e.g. the cornea and the conjunctiva, they induce an immunopathologic reaction, which can result in blindness after many years of chronic infection [6].

Current international control programs focus on the reduction of microfilaridermia by mass drug administration of ivermectin annually or biannually for more than 10–15 years. However, adult *O. volvulus* worms can live in infected humans within the nodules for over 15 years, and this entails a prolonged treatment course of ivermectin. The goal is to interrupt transmission of the parasite from human to human. Even the success achieved thus far [7–9], it must now be weighed against the fact that since 1995 only a 31% reduction in the incidence of onchocerciasis has been achieved in Africa [10]. Optimists call for additional 1.15 billion treatments to achieve elimination by 2045 [1]. Mathematical modelling and expert opinions are more pessimistic, indicating that onchocerciasis in Africa cannot be eliminated solely through mass drug administration with ivermectin alone [11], especially as this strategy cannot be applied in 11 Central African countries that are co-endemic with *Loa loa* infections; application of ivermectin may entail the risk of severe adverse events [12,13]. In 2014, the African Programme for Onchocerciasis Control called for the development and testing of new intervention technologies [14]. An alternative to the reduction of the microfilariae load is to target its source, the encapsulated adult *O. volvulus* worms with macrofilaricidal drugs. Progress in this arena requires, first, a comprehensive understanding of the biology of *O. volvulus*-human

host interactions and, second, access to adult worms for screening and/or confirmatory testing [15]. Yet, access to adult worms is challenging, as humans remain the only definitive host for *O. volvulus* and no small-animal models exist for propagating the parasite to its adult stages. Thus, the adult worms can only be obtained surgically from subcutaneous nodules, thereby limiting their availability for investigational and translational research. As a consequence, drug discovery studies are mainly performed using other filarial parasites that can be maintained in small animal models [16]. To overcome this limitation, effective novel human-derived model systems for *O. volvulus* are essential. Such models will not only allow access to resources for *in vitro* testing of large numbers of worms with drugs but also enable studying the biology of *O. volvulus* including development, nodule formation, and host-parasite interaction, all of which can contribute to a deeper understanding of this debilitating parasite and identify new tools for its elimination.

Traditional 2-dimensional (2-D) culture systems were shown to support the early stages of development for few filarial nematodes *in vitro*, with some of these parasitic worms the cultures could be maintained long-term. The reported *in vitro* survival rates ranged from 33% after 77 days for *Mansonella perstans* L3 [17], 69% after 30 days for *Brugia malayi* L3 [18], and 60–90% for *Loa loa* L3 over 15 to 17.8 days in culture [19,20]. Recently, we described a novel 2-D *in vitro* culturing system based on a feeder layer of human umbilical vein endothelial cells (HUVEC) that has successfully supported the development of *O. volvulus* fourth stage larvae (L4) to the early pre-adult stage (i.e. juvenile adults) once they have molted to the fifth-stage larvae, L5 [15]. Regrettably, the worms did not develop to adult stages as the length of a 3-month old male or female worm excised from human nodules was reported to be about 10- to 50-times longer [21]. To overcome these drawbacks of *in vitro* culture, more complex animal models were investigated [22]. However, when *O. volvulus* L3 larvae were implanted in diffusion chambers in primate, rat, mice and jird hosts up to 63 days, no more than 15% survival rates were obtained, and all were similarly developed to L4 [22]. Even in a recently developed humanized mice model representing a highly sophisticated *in vivo* "system", only 1.4% of the inoculated larvae were recovered per mouse [23]. It appears that the *in vitro* and *in vivo* newly developed systems have not yet adequately reproduced the innate human environment required for the parasite's full development to adult worms nor for their successful survival considering the % retrieval of developed worms per initial inoculum of larvae.

We hypothesized that by using 3-D culture systems that simulate the human *in vivo* niche and incorporate the following three important factors *in vitro* would potentially improve the limited survival, growth, and development of *O. volvulus* larvae observed in the 2-D culture system (Fig 1); 1) a supporting matrix enabling a 3-D environment (Factor 1), 2) human

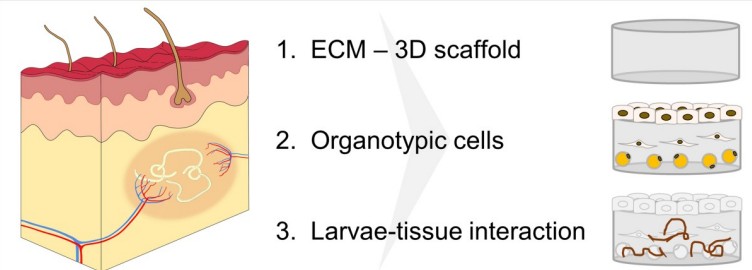

**Fig 1. Three critical factors that might support long-term *in vitro* culturing of *O. volvulus* L4s that better resemble the human *in vivo* niche, the nodule.** Factor 1: Presence of a 3-D ECM scaffold; Factor 2. Incorporation of skin-specific and subcutaneous cells with the ECM scaffold; and Factor 3. Facilitating direct contact between the larvae and skin tissue models.

niche-specific cell lineages (Factor 2), and 3) allowing a direct contact between the larvae with the human tissue equivalents (Factor 3). These three factors were not present in the optimized 2-D culture system we previously developed [15].

To provide in this study an *in vivo*-like 3-D habitat for *O. volvulus* larvae, we translated these three factors into a unique 3-D *in vitro* tissue culture system by integrating innovative tissue engineering interdisciplinary methods aimed to generate functional tissues based on scaffolds and cells relevant to the tissue of interest [24]. Engineered tissues have also been used as an alternative to animal testing, e.g. for drug development or the safety assessment of chemicals [25,26]. Furthermore, these *in vitro* tissues serve as advanced therapy medicinal products in regenerative medicine to restore defective organs and tissues [27,28]. Considering that subcutaneous tissues are the most easily palpable niche of *O. volvulus* in humans, a variety of skin equivalents with different complexities, which have already been developed, were considered for our proof of concept study; epidermal models, composed of human epidermal keratinocytes (hEK) which form a stratified epithelial layer, and which renders an equivalent for the outermost barrier of the human body [29]. These epidermal tissues can be combined with a dermal component consisting of human dermal fibroblasts (hDF) embedded in a collagen type I hydrogel forming a full-thickness skin model (FTSM). Notably, such *in vitro* skin equivalents were successfully applied to investigate the invasion process of skin-invading larvae of two parasitic worms: *Strongyloides ratti* and *Schistosoma mansoni* [30]. The presence of human cells within the skin models increased the penetration rate of the human parasite *S. mansoni* larvae, whereas the invasion by rat-specific *S. ratti* larvae was hampered. Thus, confirming that tissue-engineered *in vitro* skin equivalents can substitute the human skin for analyses of tissue-specific host-parasite interactions *in situ*. Moreover, multicellular tissue equivalents of various hosts and organs are expected to be of value for the progress of anti-parasitic prevention strategies, such as the development of drugs and vaccines [31].

In this exploratory proof of concept study, we examined the influence of three human cutaneous tissue components (Factors 1–3) on the ability to improve the fitness of long-term cultured *O. volvulus* larvae *in vitro* as compared to the traditional 2-D culture system. Fitness was evaluated based on % larval survival, motility and growth. Accordingly, we tested the impact of the FTSM consisting of a 3-D ECM scaffold (Factor 1) seeded with primary human organotypic cells (Factor 2). ECM, represents a crucial and prominent portion of native dermal tissues, generating organotypic structure and physiological properties. The infection of the human host with the infective L3 stage larvae is initiated via the vector's blood meal, followed by migration through dermal tissues to their final subcutaneous niche. Thus, we have examined the influence of a direct contact between a novel 3-D adipose tissue model with the *O. volvulus* larvae (Factor 3). Our studies were able to demonstrate that both, the indirect and/or direct co-cultures of L4s with tissue models could significantly enhance the growth of larvae in the presence of the FTSM and strongly improve the fitness as represented by the improved motility of *O. volvulus* co-cultured directly with the adipose tissue. In summary, the application of tissue engineered 3-D models represents a promising innovative tool for supporting better the long-term development of adult *O. volvulus* worms in the laboratory.

## Materials and methods

### Ethics statement

The L3 stage larvae of *O. volvulus* used in this study were produced by protocols approved by the Le Comité National d'Ethique de la Recherche pour la Santé Humaine, Yaoundé, Cameroon (Protocol 677). *Simulium damnosum* black flies were allowed to feed on consenting infected donors. The donors were offered ivermectin at the end of the 6-month duration of

their participation in the study. The infected black flies were maintained for 7 days after which they were dissected, and the L3 harvested, washed extensively, and cryopreserved before being shipped to New York Blood Center (NYBC). All procedures using these cryopreserved L3 larvae were also approved by the New York Blood Center's IRB (Protocol 321 and Protocol 603–09).

The peripheral blood mononuclear cells (PBMCs) used to culture the *O. volvulus* L3s were isolated from deidentified leukopak units collected by the New York Blood Center Component Laboratory. A written informed consent was collected from all donors. All protocols were conducted in accordance with National Institutes of Health guidelines for the care and use of human subjects.

Primary human epidermal keratinocytes (hEK) and dermal fibroblasts (hDF) used for the construction of skin models were isolated from foreskin biopsies obtained from juvenile donors aged between 1 and 3 years under informed consent. Mesenchymal stromal cells (hMSC) were isolated from bone marrow of the femur head of osteoarthritis patients undergoing surgical femur head replacement according to ethical approval granted by the institutional ethics committee of the Julius-Maximilians-University Würzburg (vote 280/18 and 182/10).

## Media

**L3 wash medium.**   This medium was prepared using 1:1 of NCTC-109 (21340039; Thermo Fisher Scientific, USA) and IMDM (31980030; Thermo Fisher Scientific, USA) with 2X antibiotic-antimycotic (15240062; Thermo Fisher Scientific, USA).

**L4 medium.**   A combination of 20% NCTC-109, 20% MEM-alpha (12571063; Thermo Fisher Scientific, USA), 20% DMEM/F12 (10565018; Thermo Fisher Scientific, USA) and 40% EGM-2 (EBM-2 Lonza CC-3156; Clonetics) formed the base medium. This medium was then supplemented with 20% heat inactivated FBS (F0926; Sigma, USA), and "1X larvae supplements", which is composed of: 1X antibiotic-antimycotic, 1% D-glucose (G8769; Sigma, USA), 1 mM sodium pyruvate (11360070; Thermo Fisher Scientific, USA), 1X insulin transferrin selenium (51500–056; Thermo Fisher Scientific, USA), 0.1% Lipid Mixture-1 (L0288; Sigma, USA), and 1X Non-Essential Amino Acids (11140–050; Thermo Fisher Scientific, USA). This medium (S1 Table) was used to co-culture L4s with the HUVEC feeder-layer.

**L4H medium.**   The L4 medium was adapted for use in the 3-D cultures by changing the percent of the basic medium components to 30%, 30%, and 40% parts of NCTC-109, MEM-alpha, and DMEM/F12, respectively, supplemented with 20% FBS and 2X larvae supplements. This modified media was diluted 1:1 with E10 (S1 Table).

**Skin model medium (E10).**   The skin models used in this study were cultured in the E10 medium which was prepared using Epilife medium (MEPI-500CA; Thermo Fisher Scientific, USA) as the base medium supplemented with 10% Fibrolife (LL-0001; Lifeline cell technology, USA), 1X Human keratinocyte growth supplement (S0015, Thermo Fisher Scientific, USA), 1.44 mM $CaCl_2$ (C7902; Sigma, USA), 10 ng/ml Keratinocyte growth factor (K1757; Sigma, USA), 252 μM ascorbic acid (A8960; Sigma, USA) and 1X Pen/Strep (P4333; Sigma, USA) (S1 Table).

**hMSC growth medium (GM).**   For the expansion of hMSCs, DMEM/F12 (31331028, Thermo Fisher Scientific, USA) was supplemented with 10% FBS (FBS.EUA.0500; Bio&Sell), 5 ng/ml hbFGF (233-FB; R&D systems, USA), and 1X Pen/Strep.

**Adipogenic differentiation medium (ADM+).**   For the differentiation of hMSCs, DMEM (61965026, Thermo Fisher Scientific, USA) was supplemented with 10% FBS, 1 μM Dexamethasone (D4902; Sigma, USA), 1 μg/ml Insulin (I9278; Sigma, USA), 100 μM Indomethacin (Sigma; I8280, USA), 500 μM IBMX (A0695; Applichem, USA), 1% D-Glucose (G8769; Sigma, USA), and 0.1% Lipid mix (L0288; Sigma, USA).

**Experimental co-culture media.** The culture media had to be adjusted and experimentally tested as it had to support harmoniously the fitness of both the FTSM, comprising of two types of primary cells (described below), and the L4 larvae. Hence, we tested 4 different media combinations using E10 medium as the base media and gradually supplemented it with the various components of the L4H medium (described above and in S1 Table): 1) 3D-1, E10 medium, 2) 3D-2, E10 + 10% FBS, 3) 3D-3, E10 + 10% FBS + 1X larvae supplements, and 4) 3D-4, E10:L4H (1:1) with a final concentration of 10% FBS and 1X larvae supplements. The 4 media combinations were compared to the L4 medium that is used routinely for the 2-D HUVEC co-cultures. As 3D-3 and 3D-4 were confirmed as the most supportive media compositions, these two media were applied for further co-culture experiments.

## Tissue model construction

**Skin tissue models.** Primary human epidermal keratinocytes (hEK) and dermal fibroblasts (hDF) were isolated from juvenile foreskin, and then expanded until passage 2 and used for the construction of FTSMs as previously described [32]. For the dermal component, trypsin treated hDF were seeded into a collagen 1 hydrogel at a density of $1x10^5$ cells/ml in DMEM (61965026; Thermo Fisher Scientific, USA) + 10% FBS (S0415, Biochrom, Germany), and the gel containing the cells was allowed to polymerize overnight. Subsequently, Accutase (A1110501; Thermo Fisher Scientific, USA) detached hEKs ($4.3x10^5$ cells/cm$^2$) were seeded on top of the dermal models. The models were then cultured at the air-liquid-interface in E10 medium to support optimal formation and stratification of the epidermis.

**Adipose tissue models.** Human bone marrow derived mesenchymal stromal cells (hMSC) were used for the construction of the 3-D adipose tissue model. hMSC were expanded in hMSC growth medium. For the generation of the models, cells were trypsinized and seeded with $1x10^5$ cells/well into a U-shaped 96-well-plate, followed by an overnight incubation in GM. Subsequently, the so formed aggregates were transferred to cell culture inserts (PIHP01250, Merck Millipore, Germany) and the growth medium was replaced with Adipogenic differentiation medium (ADM+). The aggregates were differentiated for 3 weeks. For the setup of 3-D tissue culture, the cells were used at passage 4 to 5.

## 2-D *in vitro* cultures of *O. volvulus* L3 and L4

**Preparation of *O. volvulus* larvae L4.** Various batches of cryopreserved L3s were thawed and washed with L3 wash media (described above). Each batch of L3s consists of pooled larvae that were recovered on the same day from hundreds of infected back flies and cryopreserved at the end of day. Each experiment was performed using 2–4 distinct batches of thawed L3s selected based on the expected average percentage molting and the number of L4s needed for the experiment. Accordingly, the L4s in each experiment and between the experiments listed below are from multiple biological replicas. Accordingly, L3s (10 per well) were cultured in 96-well-plates using the wash media that was supplemented with 20% FBS in the presence of 1.5 x $10^5$ PBMCs per well. The cultures were maintained at 37˚C with 5% $CO_2$ for 7–14 days after which molting from L3 to L4 stages was determined. The L4 larvae were identified by their characteristic coiling movement in addition to the presence of the shed L3 cuticle. These worms were then collected using a stereoscope and used for further culturing in the various 3-D models and in 2-D culturing system as the control.

**Identification of an experimental co-culture medium that supports skin model formation and L4 larvae development.** First, to ensure that the components within the skin model media did not have any detrimental effects on L4 development, experiments were carried out in which the four experimental co-culture media 3D-1 to 3D-4 were compared. The L4 larvae

were plated in duplicates [about 10 worms per transwell (CLS3472; Sigma, USA)] in a 24-well-plate containing $5x10^3$ HUVEC/well and divided into 5 experimental groups. The first four groups were cultured each in one of the four media: 3D-1, 3D-2, 3D-3 or 3D-4 (described above and in S1 Table). The fifth group, "2D-control" group, was cultured in L4 medium, the established 2-D culture condition for the long-term culturing of L4 [15]. The worms were observed for motility and growth over a period of a week. A Nikon DS-Fi12 camera and Nikon Eclipse TS100 inverted microscope was used to measure the length of the worms using the NIS Elements version 4.3 Windows based imaging program.

## Co-cultures of *O. volvulus* L4 and the various tissue models

**Indirect co-culture with 3-D Full-thickness skin models.** As the direct culturing of L4 within the skin models posed many technical challenges in terms of monitoring the worms (described in results), we devised an indirect co-culture setup by placing two 24-well Millicell standing inserts (PIHP01250, Millipore), one containing the FTSM and the second containing 15-day old L4 larvae (20 L4s/insert) into one well of a 6-well-plate containing the respective culture medium. Testing of the experimental co-culture media (3D-1 to -4) in the 2-D and indirect 3-D culture system confirmed that the fitness of L4 larvae was better supported by the media 3D-3 and -4. Thus, an indirect co-culture was set up using only two experimental media, 3D-3 and 3D-4. A third group of worms was cultured using the L4 medium as previously described as a 2-D control. The larvae were cultured for a period of 11 weeks. The FTSM in each well was replaced with freshly engineered skin model every 4 weeks. The models were subjected to histological analysis to ensure the viability and integrity of the model throughout the culture period. The HUVEC monolayer was replaced every week throughout the culture period. The length of the worms was measured on a weekly basis.

**Direct and indirect co-cultures of L4 with adipose tissue models.** The hMSC aggregates (N = 10) were placed into Millicell 24-well-culture inserts for the direct co-culture and on the bottom of a well of a 12-well-culture plate for the indirect co-culture. Adipogenic differentiation was initiated by culturing in ADM+ medium (described above) for 3 weeks. 14-day old L4s were added to the adipocyte aggregate containing inserts for the direct co-culture setup, and into empty culture inserts which were then placed into the adipocyte aggregate containing wells for the indirect co-culture. Co-cultures were maintained using the 3D-4 medium over a period of 5 weeks. Larvae were analyzed weekly for growth and motility using an Evos microscope (Thermo Fisher Scientific, USA).

**Scoring of larvae motility in the co-culture experiments.** For the analysis of larvae motility (% motility) a scoring system was used based on the extend of larval movement and appearance. Coiling larvae were categorized as 1; Moving larvae as category 2; slow moving, dark/discolored larvae as category 3; and dead/ not moving larvae as category 4. Categories 1 and 2 were considered as viable larvae, whereas L4 of categories 3 and 4 were considered as non-motile and dead. % motility was then calculated as the portion of category 1 and 2 larvae of the total number of worms.

## Histology

**Hematoxylin and Eosin.** For histological analysis, FTSM were washed with PBS⁻ and fixed in 4% PFA solution (P087.1; Carl Roth, Germany). The tissues were then embedded in paraffin and further processed for histological Hematoxylin and Eosin (HE) staining, followed by microscopic observation.

**Oil red O staining.** The analysis of lipid accumulations in adipocytes was performed by Oil Red O. The cells/aggregates were rinsed with PBS⁻, followed by PFA fixation. After washing

with water and incubation in 60% isopropanol, the cells/aggregates were completely covered in Oil Red O staining solution (stock solution: 0.5 g/100 ml isopropanol, staining solution: 60% stock + 40% demineralized water; O0625; Sigma, USA) followed by subsequent washing steps with 60% isopropanol and water. For the semi-quantification of lipid content, the cells were lysed by 100% isopropanol. The absorption was measured at 410 nm using a Tecan Infinite 200 plate reader (Tecan, Switzerland).

## Statistics

Data from worm growth was recorded weekly and monitored for differences between the 3-D versus 2-D conditions during the culture period and the significance of the difference in length was tested using Mann-Whitney U test for comparison of each group from start to end of culture in this system (P < 0.0001: ****) and using ANNOVA with Dunnet's analysis for comparison of the 3-D groups to the 2-D group at D93 (P ≤ 0.001: ***; P < 0.0001: ****) (Graphpad Prism Ver. 6.07).

## Results

### Survival of L4s in 2-D cultures

Recently, we described a novel 2-D *in vitro* culturing system (Fig 2A) that has successfully supported the development of *O. volvulus* L4 larvae until the early pre-adult stage [15]. Regrettably, these enhanced culture conditions did not support the survival rates of the pre-adult developed over time; generally, more than 80% of the initial L3 inoculum was lost on average based on their reduced or complete loss of motility (Fig 2B).

### Fitness of L4 cultured in the experimental co-culture media

As the ultimate goal of this study was to co-culture *O. volvulus* L4 larvae with the *in vitro* skin models, it was crucial first to ensure the ability of L4s to tolerate the additional media components of E10 that are used routinely to culture the skin models. We tested the effect of four different media combinations (3D-1 to 3D-4) using the 2-D culture system with the HUVEC feeder layer to identify the most optimal media combination that does not have detrimental effects on the larvae (N = 18–22 per group) over a period of 7 days; development and motility of the larvae. Notably, the 3D-3 and 3D-4 media allowed a better larval growth with a median length of 772.0 μm (range 649.5–858.1 μm) and 785.2 μm (range 634.9–858.3 μm), respectively, that was comparable to 761.0 μm (range 680.1–841.0 μm) in the 2-D control group (2D-

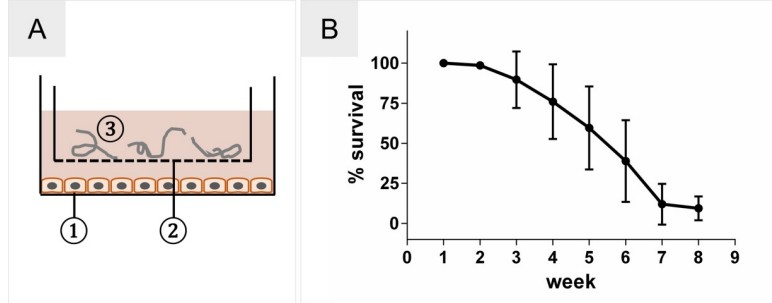

**Fig 2. Optimized 2-D culture system for *O. volvulus*.** **(A)** Schematic 2-D culture system. 1) HUVEC feeder layer; 2) culture insert with porous membrane; 3) L4 larvae loaded into the insert. **(B)** Loss of 80% of larvae during culture from L3 to early pre-adult stages over 8 weeks of culture using the 2-D culture system. The results presented are an average of 18 batchs of larvae (300–600 L3s per batch) that were cultured and monitored weekly for motility.

ctrl) (Fig 3), as well as sustained larval fitness as specified by the number of remaining highly motile L4s on day 7 (67% and 78% for 3D-3 and 3D-4, respectively, versus 76.2% for 2D-ctrl). Culturing in E10 (3D-1) or E10 supplemented with FBS only (3D-2) were not as supportive (motility of 31% and 61%, respectively, and with somewhat reduced median length; 699.4 and 754.4 μm, respectively). To confirm the results obtained from the 2-D culturing setup for the experimental media in a preliminary experiment, we also tested the four media combinations 3D-1 to 3D-4 in the indirect co-culture set up. Again, it appeared that 3D-3 and 3D-4 were still the best to support the growth of L4s (S1 Fig), and hence these two media combinations were chosen for subsequent long-term culturing of *O. volvulus* L4 larvae.

## Fitness of the 3-D models using the experimental co-culture media

The FTSM represents the two outermost layers of the human skin, the dermis and epidermis, and therefore is cultured at the air-liquid interface (Fig 4A and 4B). On top of the collagen-based dermis, keratinocytes built a functional stratified epithelium, which is generated by pro-liferating cells of the basal layer that differentiate and migrate towards the stratum corneum (Fig 4C). In order to co-culture the FTSM with *O. volvulus* larvae, which each requires its own complex culture medium, we also tested the compatibility of the tissue with the experimental co-culture media (3D-2 to 3D-4) versus the E10 (3D-1) medium (S1 Table). The histological analysis of the models cultured with 3D-2 to 3D-4 media and L4 medium (S2 Fig) revealed changes associated with the formation of the epidermal layer. These alterations are mainly presented by a shift from highly prismatic cells (3D-1 and -2) in the stratum basale towards iso-prismatic keratinocytes (3D-3, -4 and L4 medium). Furthermore, especially in the stratum spinosum a vacuolization of the cells occurs. In contrast, the dermis did not show any differences when cultured with media 3D-3 and 3D-4 (Fig 4D and 4E). Although the epidermal layer was not fully intact compared to the E10 medium condition, we elected to continue using

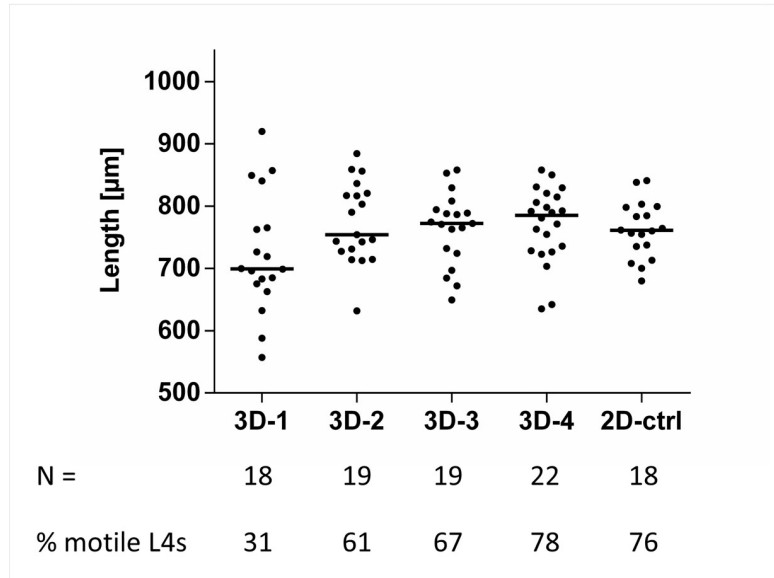

**Fig 3. Compatibility of the experimental co-culture media on L4 fitness using the 2-D culturing setup with a HUVEC monolayer.** Growth (length, median per group) of larvae and % of L4 of the initial inoculum (N = 18–22 per group divided in two reproducible wells) that were still motile in the presence of experimental co-culture media (3D-1 to 3D-4) in comparison to the 2-D culturing system (2D-ctrl) after 7 days in culture. The number of L4s per group and the % motile L4s in each group of day 7 in culture are indicated below the graph.

**Fig 4. Structural analysis of FTSM cultured in the experimental co-culture media. (A)** Schematic of the FTSM comprising epidermis and dermis cultured in **(B)** cell culture insert on a porous membrane. **(C)** Histological Hematoxylin and Eosin staining of FTSM in E10 skin medium (3D-1) compared to **(D)** 3D-3 and **(E)** 3D-4, the best experimental co-culture media that supported L4 fitness. Scale bar = 100 μm. (1) porous membrane, (2) outer-well medium, (3) dermis, (4) epidermis.

the FTSM with these media through all the subsequent experiments as it did not appear to affect the fitness of the larvae negatively (Fig 3).

## Direct culture of larvae within the collagen-based skin models

In a first step and based on the hypothesis of this study, *O. volvulus* larvae were cultured in direct contact with the ECM and cells within the 3-D skin model. Different approaches were tested for their feasibility to introduce and culture larvae in the collagen-based dermal model (Fig 5). Similar to the *in vivo* infection route of *O. volvulus* L3, a few strategies for inoculation were tested. First, we injected L4 larvae into the dermis (Fig 5A), and as second strategy, a wound created by a tissue punch was inoculated with the larvae (Fig 5B). The L4s inside the wound were monitored using light microscopy (Fig 5C) for growth and motility. Regrettably, the larvae migrated into the tissue through the undefined wound edges into the collagen and were not traceable thereafter. The same challenge occurred when larvae were injected into the dermis, or when we tried to directly add the L4 larvae into the collagen scaffold during the setup of the model (Fig 5D). A more advanced strategy was then developed in which a confined compartment was imprinted into the dermal tissue model composed of collagen with hDF using a stamp specially constructed for this purpose; the newly created compartment was then loaded with 15-day old L4s (Fig 5E and 5F). This novel setup enabled a direct contact between the 3-D model and the larvae while keeping them within the confined compartment (S1 Video). Notably, the larvae grew significantly over time and after day 28 in culture the median length was 786.4 μm (range 701.4–875.8 μm) as expected (Fig 5G). Unfortunately, due to the continuous remodeling of the collagen scaffold by the hDF the compartment collapsed (S3 Fig), resulting in the disappearance of a significant number of the worms (reduction of 44%) within the scaffold (Fig 5G), which impeded the L4 monitoring and hence the long-term culturing of the larvae in direct contact with the skin models. Moreover, in the 3D culture system, the larvae are present in multiple layers, which makes it also more difficult to monitor their length accurately.

## Optimized indirect co-culturing of larvae with skin models

Faced with challenges to monitor and track the larvae using the direct culturing within the skin models, we proceeded with an indirect setup of skin tissue models and larvae, which still allowed us to explore the augmenting effect of soluble factor exchange on the fitness of the larvae. A spatial separation was generated by using two standing cell culture inserts, one that contained the model and the other that contained the larvae while still enabling their communication by factors secreted into the culture medium through the porous membranes (Fig 6A and 6B). We were able to maintain this indirect co-culture system up to 11 weeks using the experimental co-culture media we already selected (3D-3 and 3D-4) in previous experiments (Figs 3 and 4). Weekly measurements of the larvae length demonstrated that both

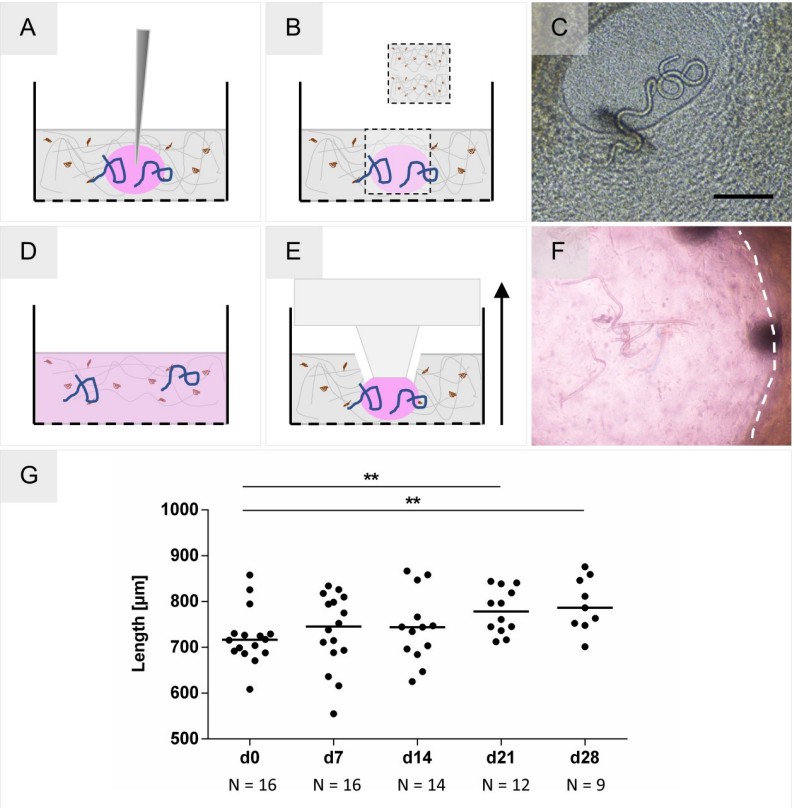

**Fig 5. Methods for introducing larvae into a collagen-based tissue model. (A)** Injection of larvae. **(B)** Creation of a wound with a tissue punch, followed by loading of larvae. **(C)** Microscopy of L4 inside the undefined wound created within the 3-D model. Scale bar = 200 μm. **(D)** Homogenous distribution of larvae by their incorporation into the collagen during model creation. **(E)** Generation of a confined compartment by imprinting a well with a stamp. **(F)** L4 larvae inside the imprinted compartment within the dermal model on day 2 of culture. The image was taken using light microscopy through the insert membrane and a thin collagen layer, responsible for the blur. **(G)** Growth of L4 (N = 16 divided between two reproducible tissue models) cultured in direct contact within imprinted compartment in the 3-D models on day 7, 14, 21, and 28 in culture. The significance of differences in growth between D0 and D21 (P = 0.0061: **) and D0 and D28 (P = 0.0065: **) was analyzed using the Mann-Whitney U test. The number of worms, that could be observed is noted below the X-axis.

experimental co-culture media supported a significant growth of L4 compared to the 2-D culture condition. On Day 77 (Worm age D92) of culture the median length of 816.3 μm (3D-4; range 744.4–934.8 μm) and 830.8 μm (3D-3; range 765.1–897 μm) was obtained versus 766.6 μm (range 634.8–836.6 μm) in the 2-D culture (Fig 6D). However, the 3D-3 media was not as supportive as 3D-4 in maintaining the motility of the developing larvae; 2-fold more larvae were motile in 3D-4 on day 77 of culture. Interestingly, a new observation was made; fibroblasts from the FTSM migrated towards the culture insert containing the larvae (Fig 6A), which resulted in the integration of the L4 into a newly formed cluster of hDF and ECM within their own insert (Fig 6C). This has been observed as early as day 21 of culture and without affecting the fitness of the larvae over time.

## Development and optimization of the adipose tissue model for co-culturing with L4 larvae

We developed a novel adipose tissue model for the co-culture of *O. volvulus* larvae as a representation of the underlying subcutaneous tissues where the nodules reside in infected

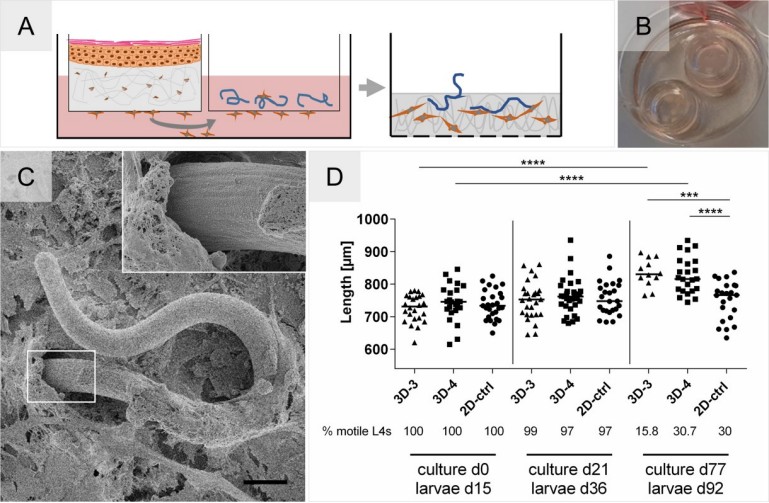

**Fig 6. Indirect 3-D co-culture setup that supports the growth of *O. volvulus* L4. (A)** FTSM and larvae are co-cultured in separate culture inserts with a schematic representation of hDF migration into the larvae compartment. **(B)** Image of the two inserts in a 6-well-culture plate containing the 3-D indirect co-culture setup. **(C)** SEM picture depicting cells that migrated inside the larvae containing insert and grew over the viable *O. volvulus* L4. Inset shows the interface between L4, cells, and ECM. Scale bar = 20 μm. **(D)** Growth of 15-days old L4 co-cultured in the indirect 3-D setup using two experimental co-culture media, 3D-3 and 3D-4, versus 2D-ctrl (over a HUVEC monolayer) over 77 days of culture (observations reported are those for day 15, 36, and 92 of worm age). Each experimental condition was set up with 3 technical replicates containing ~10 L4s each. The % motile L4s in each group and time point are indicated below the graph. The significance of differences in growth between days in culture was analyzed using the Mann-Whitney U test (P < 0.0001: ****), and between the groups on day 77 of culture using ANNOVA with Dunnet's analysis (P ≤ 0.001: ***; P < 0.0001: ****)

individuals; the adipose tissue model is based on hMSC aggregates grown within the cell culture inserts (Fig 7A). Notably, the accumulation of triglycerides inside intracellular lipid

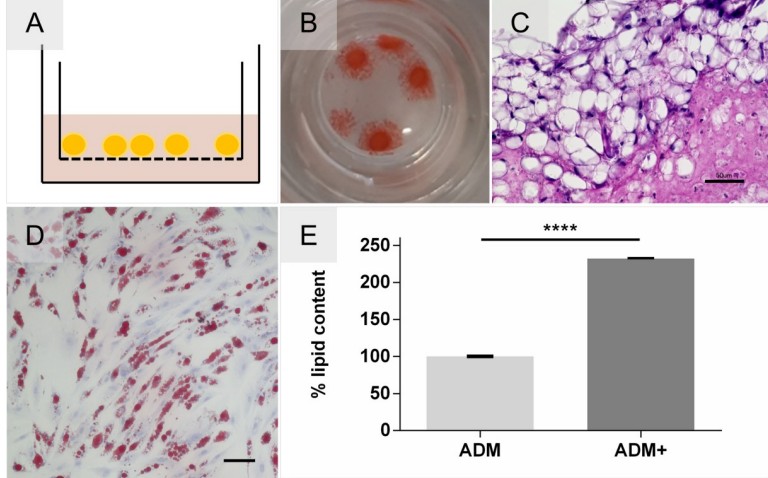

**Fig 7. Development of an adipose tissue model with an enhanced differentiation efficiency. (A)** Schematic Adipose tissue model cultured on cell culture insert membrane. **(B)** Adipocyte aggregates stained positive for Oil Red O. **(C)** Histological Hematoxylin and Eosin staining depicts formation of adipose tissue-like organization containing mature uniocular adipocytes. Scale bar = 50 μm. **(D)** Tolerance testing of hMSCs cultured in 2D and differentiated in L4 medium supplemented with adipogenic differentiation factors, confirmed by Oil Red O. Scale bar = 100 μm. **(E)** Oil Red O Quantification of enhanced lipid accumulation by supplemented glucose and lipids (ADM+) compared and normalized to standard differentiation medium (ADM). Unpaired t-test P < 0.0001, N = 2.

vacuoles was evident and confirmed by a positive Oil Red O staining and by the typical mature adipocyte phenotype showing a unilocular vacuole in histological analysis (Fig 7B and 7C). Importantly, we also analyzed the tolerance of the adipose tissue to the composition of the L4 media. We found that hMSCs not only differentiate adipogenically in L4 medium supplemented with adipose differentiation factors but that their potential to accumulate lipids was further enhanced by the additional glucose and lipids present in the L4 medium. (Fig 7D and 7E). Due to this observation, the final adipogenic differentiation medium used for this study is supplemented with additional lipids and glucose (ADM+).

## Co-culturing of larvae with adipose tissue models

To investigate the benefit of a physical contact between L4, cells and matrix, the influence of the *O. volvulus* niche-specific adipose tissue model on larval fitness was analyzed using two distinct setups: via soluble factors versus a direct contact. Accordingly, larvae were cultured within culture inserts with or without contact with the adipocyte aggregates (Fig 8A and 8B). As shown in Fig 8C, the larvae were in direct proximity to the aggregates in the direct contact setup and attached to the undifferentiated hMSCs which covered most of the culture insert membrane. Like the migratory overgrowth in the soluble FTSM co-cultures (Fig 6C), no detrimental effect was observed on these directly cultured L4 (S2 Video). Conversely, a strong positive effect on the fitness of the larvae was observed by an increase in motility of 69% (absolute 95%) in the direct co-culture versus 11% (absolute 37%) in the indirect co-culture that was the same as observed for L4 cultured in the 2-D control system, which on day 52 had only 26% of motile larvae (Fig 8D). We have not measured the differences in the length of the L4s cultured within the inserts with or without contact with the adipocyte aggregates as the larvae do not

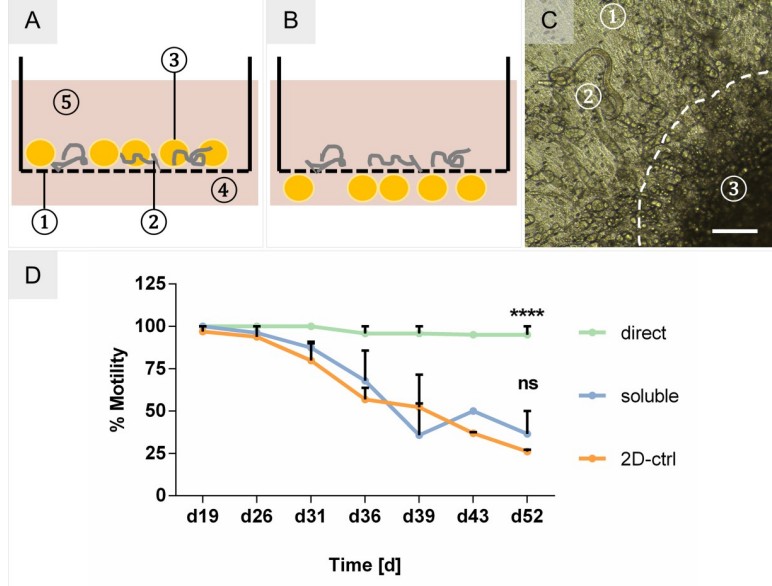

**Fig 8. Adipose tissue co-culture setups. (A)** Schematic of direct adipose aggregate co-culture with *O. volvulus* larvae. 1. culture insert with porous membrane; 2. L4 larvae; 3. adipose aggregates; 4. outer well; 5. culture medium. **(B)** Indirect co-culture setup of larvae and adipose tissues. **(C)** Microscopic picture of L4 with adjacent adipose tissue aggregate, indicated by dashed line. 1. Porous membrane; 2. L4 larva; 3. adipose aggregate. Scale bar = 200 μm. **(D)** % Motility of L4 co-cultured in the 2-D culture system (2D-ctrl), over adipose aggregates in indirect contact (soluble), and in direct contact with the adipose aggregates (direct). Two-way ANOVA–Significance tested against the 2-D culture system as the control. 2D-ctrl: N = 27 L4; soluble: N = 33 L4; direct: N = 30 L4.

grow in length much until day 60–70 of their development. The major increase in length occurs after day 70 in culture when they have molting to L5 [15].

A complete summary of various conditions tested, and their outcomes can be found in S2 Table.

## Discussion

In this proof of concept study, we aimed to identify whether three critical factors present in the human *in vivo* nodular niche (Factors 1–3) where the *O. volvulus* worms develop and reside can support the long-term culturing *in vitro* of *O. volvulus* L4s. The three factors are: a supporting matrix enabling a 3-D environment (Factor 1); cell-specific lineages incorporated into the scaffold that present in the nodular niche (Factor 2), and culturing the larvae in direct contact with the human tissue equivalents (Factor 3). Our hypothesis was that by adapting these three organoid factors into *in vitro* culturing systems would overcome some the drawbacks of the traditional 2-D cultures used for filarial parasites. These 2-D culture systems typically contain a feeder-layer made of a single cell type that does not necessarily represent the complex 3-D natural niche and/or deliver the appropriate host stimuli required by the parasite. Even though we reported the successful *in vitro* development of *O. volvulus* to pre-adult worms using the 2-D culturing system with HUVEC as the feeder layer [15], the survival of the larvae over time was low (Fig 2). Viability drops in the first 7 weeks very rapidly. In this pilot study, we reconstructed in part the *in vivo* nodular niche by applying engineered *in vitro* tissue culture systems enabling a 3-D organization of skin-specific cells that might better support a natural interaction with the developing larvae (Fig 1). The study focused on the first 7 weeks, when usually the drop of viability is usually detected.

The beneficial impact soluble factors secreted by cells have on the growth and development of *O. volvulus* larvae was already observed using 2-D culture systems with feeder layers of HUVEC and human dermal fibroblast cell line [15]. Although both cell lines supported the growth of the L4s to pre-adult worms, the use of a human dermal fibroblast cell line in 2-D cultures was logistically challenging as the cells overgrew within the culture plates causing the growing larvae to become entangled within the cellular monolayer, which subsequently affected negatively on their fitness. HUVEC on the other hand do not overgrow in culture. Our indirect co-culture system with the FTSM further confirmed the favorable impact of soluble factors secreted by the niche-specific cells cultured within the ECM scaffold; the growth of the larvae was significantly improved with median length of 831 μm (8.4% increase, 3D-3) and 816 μm (6.5% increase, 3D-4) versus 767 μm in the 2-D system (Fig 6). We believe that the increased culture complexity provided by the 3-D collagen scaffold (Factor 1) seeded with two skin-specific cell types (Factor 2) may have driven the improved growth of the L4s by augmenting the culture medium with unique human physiological signatures that can otherwise be hard to reproduce empirically. The identification of such tissue-secreted factors to create a customized medium would be very time-consuming and cost prohibitive. In the co-culture system, the release of these factors is possibly a rather dynamic process that results from the communication between the worms and the tissue, which cannot be substituted by classical media exchanges at defined intervals in a conventional culture system.

Due to the different components of the 3-D co-culture system including the multiple cell types and the larvae, it was crucial to first identify an optimal medium combination that keeps all the elements in this unique system nourished harmoniously through the long-term culture. Testing the four experimental co-culture media, using the skin model medium E10 (3D-1) as the base medium, has determined that media 3D-3 and 3D-4, both of which contained the larvae media and/or supplements, seemed to be more favorable for the growth and motility of the

*O. volvulus* L4; 3D-1 and 3D-2 did not support larval fitness to the same degree (Fig 3 and S1 Fig). These findings signified the crucial role of the supplements (glucose, sodium pyruvate, non-essential amino acids, and lipids) for the fitness of the worms regardless of the culture setup. In the FTSM, the differentiation of keratinocytes and the stratification of the epidermis is a rather sensitive process, vulnerable to changes in the culture medium. The differentiation and migration of keratinocytes from the stratum basale towards the stratum corneum was impaired when cultured with the experimental culture media (3D-2 to 3D-4 and L4 medium) versus E10 (3D-1), whereas the dermis was not affected (S2 Fig). Nonetheless, the FTSM co-cultured with larvae in 3D-3 and 3D-4 still contributed to their growth (Fig 6D), raising the possibility that the epidermal layer does not have any significant impact for *O. volvulus* development *in vivo* and *in vitro*.

Based on the natural location of *O. volvulus* larvae *in vivo*, we assumed that the most logical strategy would be to co-culture larvae within the 3-D *in vitro* tissues, facilitating their direct contact and interaction with the cells and ECM (Factor 3). Regrettably, although the worms were maintained within the imprinted models and demonstrated growth (Fig 5F and 5G) comparable to the indirect co-culture system (Fig 6D), this endeavor was hindered by the limited optical features of the collagen-based dermal tissue models; inhibiting robust monitoring of the worms. Confronted with these technical challenges (Fig 5A–5F), we developed the alternate indirect co-culture system in which we were able to culture L4 larvae until they were 92 days old and already developed by then to the pre-adult stage [15]. Like the 2-D culture system, the larvae were kept in the cell culture inserts separately, which enabled us to monitor them by light microscopy. However, in this setup, the insert containing the larvae was placed together with the second insert containing the FTSM within the same tissue culture well (Fig 6A and 6B), thereby still enabling the bi-directional communication via soluble factors. Interestingly, this indirect co-culture with L4 resulted in a strong migration of hDF from the tissue model towards the larvae and the formation of ECM, which possibly confirms active crosstalk between the larvae and the tissue model. In our previous 2-D studies [15] investigating the possibility that hDF migrate actively towards the larvae was hampered by the overgrowth of the monolayer with time. Future studies will establish whether a hDF monolayer alone could be used to support the culturing of the larvae if their overgrowth with time is well controlled in comparison to the FTSM system. Interestingly, as active fibroblasts were identified at the inner edge of the fibrous capsule of the human onchocercoma [33], further studies using our novel tissue models might support a better understanding of the role fibroblasts have during nodule formation. We wonder if the recruitment of cells from the skin tissue model towards the larvae could possibly be an initial step of triggering nodule formation by the *O. volvulus* worms. Like the nodular niche, *O. volvulus* larvae were seen to be integrated into the formed cluster of cells and ECM secreted by the hDF, as demonstrated by the SEM (Fig 6C). Moreover, subcutaneous nodules also contain various immune cells, embedded in fibrous tissue and supplied by vasculature [5,34,35]. Typically, a foreign body is encapsulated in fibrous tissue after its detection by macrophages *in vivo* followed by resident macrophages stimulating fibroblasts in the proximity of the foreign body to produce matrix proteins for capsule formation [36]. Surprisingly, in this study the larvae appeared to elicit migration and matrix production directly without the presence of immune cells. This finding is a hint that fibroblasts have a crucial role in the culture of the larvae and can strongly interact with the parasite even without an immune system that usually modulates the host response. Importantly, the integration of the L4 into a newly formed cluster of hDF and ECM within their own insert did not have a negative impact on larval fitness, as shown by their sustained motility. This evidently strengthens our hypothesis that the direct contact of larvae with cells and ECM is a crucial factor for the *O. volvulus* development, also *in vitro*. To further study whether this interaction is beneficial, we will need to

overcome the challenges we had with the direct contact co-cultures. Particularly, advanced microscopy methods need to be established and applied to enable non-invasive monitoring of larval fitness during the long-term cultures versus determining the vitality and growth of larvae using assays end-point assays such as the digestion of the collagen scaffold followed by MTT (3-(4,5-dimethylthiazol-2-yl)-2,5-diphenyltetrazolium bromide) assay.

Nodules containing adult *O. volvulus* worms are predominantly found in the subcutis, which mainly consists of adipose tissue [4,21,37]. To render this niche-specific tissue, we developed an adipose tissue model based on aggregated hMSCs that were adipogenically differentiated using the optimized medium for differentiation (ADM+) and the previously evaluated 3D-4 medium for larvae co-culture. Histological analysis of this model showed the typical morphology of adipose tissue with densely packed mature adipocytes. Those were well maintained in the 3D-4 medium without any deleterious effects on cell fate stability (Fig 7). In the presence of an adipose tissue model, the motility of L4 was increased by 11% in the indirect culture setup compared to the 2-D culture. In contrast to the direct culture in the skin models, the adipose tissue models consisting of cell aggregates placed on the permeable membrane of the culture insert allowed the visual observation of the L4 cultured in a direct contact with the adipocytes by light microscopy (Fig 8C). Notably, this direct contact setup improved the motility of L4 larvae by 69% after 38 days of the co-culture (Fig 8A). Similar to the embedding of L4 into the newly formed cell clusters after hDF migration and the formation of ECM in the indirect co-culture, the larvae in the direct co-culture with adipocytes were attached to the non-differentiated hMSCs in the proximity of adipose aggregates (Fig 8C and S2 Video). Thus, this co-culture experiment seems to confirm the beneficial impact of the direct interaction between larvae and tissues (Factor 3) on motility and the fitness of *O. volvulus in vitro*. Notably, *O. volvulus* worms as well as many other nematodes express fatty acid binding proteins that are essential for their fitness [38,39], and cholesterol was already shown to be a crucial factor for nematode development [40]. The possible increased availability of lipids, fatty acids and cholesterol secreted by the adipocytes and/or released by the dead cells (S3 Video) contributing to the fitness of the developing larvae when they are cultured directly versus indirectly will need more studies.

Taken together, this proof of concept exploratory study has identified crucial factors within the *O. volvulus* niche that could be translated into an optimized co-culture system utilizing 3-D tissues that benefit to larval fitness *in vitro*. The co-culturing with FTSM increased significantly the growth of L4 larvae probably via secreted factors, whereas the adipose tissue in direct contact with the larvae enhanced their motility and fitness also significantly compared to the 2-D culture system. The increased complexity and host-specificity possibly facilitated a variety of stimuli on the developing L4 larvae, which are absent in a single-cell-type-based 2-D culture. Although, a clear benefit of tissue-engineered skin models was shown, these co-culture experiments do not yet represent methodologically sound culture systems for *O. volvulus*. Nevertheless, our findings revealed various experimental aspects that will be of value as a basis for future studies aimed to develop 3-D skin tissue-based culture systems.

Accordingly, future studies should focus on establishing a direct co-culture setup mimicking the natural *in vivo* niche of *O. volvulus* adult worms more comprehensively; culturing the larvae within the FTSM and with an integrated subcutaneous adipose tissue that hopefully will further support the formation of a nodule-like structure *in vitro*, a glimpse of which we observed in the indirect co-culture system (Fig 6C). We hypothesize that by increasing the complexity of the *in vitro* 3-D culturing system will not only achieve a synergetic effect of the FTSM (growth) and the adipose tissue (viability) on larval fitness, but also when combined with direct co-culturing conditions that support monitoring over a prolonged culture period would also potentially enhance further the survival and development of L4 to the pre-adult

juvenile worms (as already done) and hopefully also to the more mature adult worms. Analyses of such developing *O. volvulus* worms for morphology, transcriptome and proteome profiling will validate their stage of development when compared to these profiles in adult worms recovered from infected patients [15,41].

Although we did not initially consider in this proof of concept study the indirect co-culture setup as an optimal culture condition for *O. volvulus*, we discovered that this culture system could be still a valuable method to unraveling the molecular basis that support the formation of the nodule and the interactions between the parasite and the distinct human host tissues within the nodule. For example, we could use a simplified and yet advanced experimental designs to establish the individual impact each of the tissue components (e.g. adipose tissue, dermis, and epidermis, varied but defined immune cells) have on larval fitness and development through the two molts that happen in the human host. We could possibly start the *in vitro* varied culturing conditions by initiating the culturing with the L3s and compare to the development of the L4s versus when the complexed co-culturing that starts with the L4s; such conditions could resemble more of the parasite's lifecycle. Furthermore, this could be easily combined with the culturing system with other supplementary factors to reflect host responses to the parasite infection on a molecular level [6]. hDF migration (Fig 6A–6C) was an unexpected albeit interesting observation that could possibly aid in better understanding of the molecular mechanism(s) that the parasite triggers to initiate nodule formation with and without the help of immune cells and/or human ECM factors. Also, it would be of interest to analyze and compare the secreted factors from the FTSM, adipose tissue and/or combined tissues when cultured with live L4s versus killed L4s to validate if the soluble factors/molecules (i.e. cytokines, chemokines, growth factors etc.) are different and thus in response to the presence of actively growing larvae. Our human *in vitro* tissue system, therefore, can enable novel opportunities to answer host-parasite interaction questions that cannot otherwise be studied in the human host or in the existing animal models. The indirect co-culture also increases the retrieval efficiency of the developed worms as compared to the complex retrieval and limitation when the humanized mice model is used [23].

Altogether, we believe that ultimately and when further optimized the *in vivo*-like tissue equivalents could provide a more comprehensive system for the generation of well-developed pre-adult and adult *O. volvulus* worms. This could also provide the filarial research community with a novel and robust source of adult worms to be used for drug screening when needed. Replacing the presently used surrogate filarial parasites for drug screening and validation *in vivo* with the target parasite for the development of novel macrofilaricidal drugs, *O. volvulus*, will also reduce the need of animal models for confirmatory testing of drug effects [16]. We assume that our *in vitro* tissues will be also applicable to studies with other filarial parasites and human helminths, contributing to the progress of anti-parasitic treatment strategies for various helminth infections [31]. The novel *in vitro* culturing system of *O. volvulus* could ultimately further support the better understanding of the biology of this obligate human parasite and the development of new interventional tools to fight this debilitating human disease.

## Supporting information

**S1 Fig. A comparison of growth after 7 days in indirect co-culture conditions.** 3D-3 and 3D-4 demonstrated better growth than 3D-1 and 3D-2.
(TIF)

**S2 Fig. Co-culture media testing on FTSM.** Hematoxylin and Eosin staining of FTSM cultured with experimental co-culture media. **A.** 3D-1 (E10), **B.** 3D-2, **C.** 3D-3, **D.** 3D-4. **E.** L4 medium. The tested media did not have a negative impact on dermal formation. All

epidermis-specific layers stratum basale, spinosum, granulosum and corneum were formed. Compared to FTSM cultured with skin model specific E10 media, the epidermal tissues cultured with 3D-2 to 3D-4 and L4 medium shows changes in the cell layer formation, especially by vacuolization in the stratum spinosum. The lack of the stratum corneum can be explained by shearing off through histological processing.
(TIF)

**S3 Fig. Direct co-culture of L4 in imprinted compartment within the dermal model.** By day 35, due to the constant remodeling of the collagen by the hDF, the model contracted causing the imprinted compartment to collapse. The hydrogel loosened from the insert wall (dashed black line) and decreased in volume, shown by the spotted black line. The imprinted compartment containing the larvae shrank (dashed white line), impeding their observation. Magnification: 2X.
(TIF)

**S1 Table. Detailed media recipes used for 2-D and 3-D cultures.**
(XLSX)

**S2 Table. A summary of the experiment goals and the outcomes of the various preliminary studies in which we optimized culturing condition that support the growth and fitness of *O. volvulus* larvae cultured in vitro.**
(DOCX)

**S1 Video. Early co-culture of L4 larvae in imprinted well within the collagen scaffold.** Viable larvae were traceable by light microscopy before contraction of collagen well.
(MP4)

**S2 Video. L4 larvae cultured in direct contact with adipose tissue aggregates.** Undifferentiated hMSCs attached to larvae in direct proximity to adipose tissue model on cell culture insert membrane.
(AVI)

**S3 Video. Secreted lipids in supernatant.** On the surface of the supernatant, an accumulation of lipid acids was found 3 days after last medium exchange. Therefore, the adipose tissue was matured for 28 days.
(MP4)

**S1 Text. List of abbreviations used within the manuscript.**
(DOCX)

## Author Contributions

**Conceptualization:** Christoph Malkmus, Shabnam Jawahar, Sara Lustigman, Jan Hansmann.

**Data curation:** Christoph Malkmus, Shabnam Jawahar, Nancy Tricoche.

**Formal analysis:** Christoph Malkmus, Shabnam Jawahar, Nancy Tricoche, Sara Lustigman, Jan Hansmann.

**Funding acquisition:** Sara Lustigman, Jan Hansmann.

**Investigation:** Christoph Malkmus, Shabnam Jawahar, Nancy Tricoche.

**Methodology:** Christoph Malkmus, Shabnam Jawahar, Nancy Tricoche, Sara Lustigman, Jan Hansmann.

**Project administration:** Sara Lustigman, Jan Hansmann.

**Resources:** Sara Lustigman, Jan Hansmann.

**Software:** Christoph Malkmus, Shabnam Jawahar.

**Supervision:** Sara Lustigman, Jan Hansmann.

**Validation:** Christoph Malkmus, Shabnam Jawahar, Nancy Tricoche, Sara Lustigman.

**Visualization:** Christoph Malkmus, Shabnam Jawahar.

**Writing – original draft:** Christoph Malkmus, Shabnam Jawahar, Sara Lustigman, Jan Hansmann.

**Writing – review & editing:** Christoph Malkmus, Shabnam Jawahar, Nancy Tricoche, Sara Lustigman, Jan Hansmann.

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
