## [Decision Letter · Decision Letter 0]

23 Jul 2020

Dear Dr. Lustigman,

Thank you very much for submitting your manuscript "Optimizing 3-dimensional human skin models to facilitate long-term development of Onchocerca volvulus, a debilitating obligatory human parasite" for consideration at PLOS Neglected Tropical Diseases. As with all papers reviewed by the journal, your manuscript was reviewed by members of the editorial board and by several independent reviewers. In light of the reviews (below this email), we would like to invite the resubmission of a significantly-revised version that takes into account the reviewers' comments. 

Reviewers agree that this work is important for the field and represent a significant technical step forward, but the are some issues that need to be addressed before we can accept this paper for publication. Please address the comments of all the reviewers, particularly the technical ones from reviewers 1 and 3 and the discussion points from Reviewer 2.

We cannot make any decision about publication until we have seen the revised manuscript and your response to the reviewers' comments. Your revised manuscript is also likely to be sent to reviewers for further evaluation.

Sincerely,

Keke Fairfax, PhD

Deputy Editor

Keke Fairfax

Deputy Editor

Reviewers agree that this work is important for the field and represent a significant technical step forward, but the are some issues that need to be addressed before we can accept this paper for publication. Please address the comments of all the reviewers, particularly the technical ones from reviewers 1 and 3 and the discussion points from Reviewer 2.

Reviewer's Responses to Questions

**Key Review Criteria Required for Acceptance?**

**Methods**

-Are the objectives of the study clearly articulated with a clear testable hypothesis stated?

-Is the study design appropriate to address the stated objectives?

-Is the population clearly described and appropriate for the hypothesis being tested?

-Is the sample size sufficient to ensure adequate power to address the hypothesis being tested?

-Were correct statistical analysis used to support conclusions?

-Are there concerns about ethical or regulatory requirements being met?

Reviewer #1: no comments

Reviewer #2: yes, with restriction, see my comments

Reviewer #3: - Clearer description of what defines biological reps would be helpful throughout. It's unclear how parasites are sourced, cryopreserved, and thawed, with respect to potential batch effects.

- It's unclear in some places (e.g., Figure 8D) how many worms are being grouped in the summary data and whether these are technical replicates from the same batch or biological replicates from different thawing and molting events. In recognition of the difficulties in phasing experiments with 'true' biological replicates, I am only suggesting that this information be included - not necessarily suggesting added replication.

**Results**

-Does the analysis presented match the analysis plan?

-Are the results clearly and completely presented?

-Are the figures (Tables, Images) of sufficient quality for clarity?

Reviewer #1: See Summary and General Comments

Reviewer #2: yes, with restriction, see my comments

Reviewer #3: - Figure 1 and 2 can be combined.

- Figure 2B: how many worms per well?

- Figure 4 would be helped by internal labels.

- Figure 5G: the d7 worm population includes at least one worm smaller than any initially measured (d0). Is this a dying worm, or does it reflect the difficulty in accurately measuring worm lengths?

- L442: related to point above, could disappearing worms be dying, and not just moving through the scaffold?

- There are two Figure 7s uploaded

- It would be interesting to speculate (and eventually know) if much of the difference between direct and indirect co-culture motility in the adipose tissue model are a result of cholesterol / lipid availability, perhaps even from dying cells. 

- A single summary table of all results (culture conditions, timelines, and mean lengths or normalized motility) would be helpful, even as a supplemental table.

**Conclusions**

-Are the conclusions supported by the data presented?

-Are the limitations of analysis clearly described?

-Do the authors discuss how these data can be helpful to advance our understanding of the topic under study?

-Is public health relevance addressed?

Reviewer #1: See Summary and General Comments

Reviewer #2: yes, with restriction, see my comments

Reviewer #3: - A shorter version of much of the conclusion is warranted, given modest success in some areas and the supplanting of one approach with the next throughout. Most attention should be given to the culture approach that shows greatest promise (direct co-culture with adipose cells).

**Editorial and Data Presentation Modifications?**

Reviewer #1: (No Response)

Reviewer #2: Minor Revision

Reviewer #3: (No Response)

**Summary and General Comments**

Reviewer #1: Malkmus et al. presented two 3-dimensional in vitro culture that allow long-term culturing of Onchocerca volvulus and facilitate development of L4 larvae to young adults. The results shown here are very interesting and crucial for the filarial research community, since a source of O. volvulus worms is still missing but is indispensable for research about the parasite’s biology and development of macrofilaricidal drugs. In my opinion, the manuscript is suitable for publication in PLOS NTD, but a couple of points need to be clarified beforehand.

Major comments:

1) Did the authors try to directly use and implement the L3 larvae into the culture systems? This would allow a continuous observation within one culture system.

2) In general, why did the authors stop the cultures after 77days (collagen-based tissue model) and 52 days (adipose tissue model), respectively? Did the motility/survival reduce after these time-points? Interestingly, Figure 8D showed that the motility in the direct adipose tissue model remain high and constant upon 52 days compared to the start of the culture.

3) In regards to the question in point 2), can the authors show the motility and survival rates for Figure 6D and length and survival rate for Figure 8D?

4) Since the authors state that fibroblasts from the FTSM migrate towards the insert containing the larvae leading to cluster formation, I am wondering if fibroblasts alone could be used for good culturing condition without the whole FTSM. Indeed, the authors discussed that tissue components need to be tested separately and questioned the presence of the epidermal layer for O. volvulus development. In addition, the connection between fibroblasts and nodule formation have been already shown, it would be really interesting and important for the message/conclusion of the manuscript, if the authors could state/discuss preliminary results or hinds that fibroblasts alone might be beneficial for O. volvulus development and maybe also nodule formation.

5) Since the parasites did not get in contact with the tissue/cells (except the fibroblasts) from the FTSM, did the authors measured soluble factors/molecules (cytokines, chemokines, growth factors etc.), which might be secreted from the FTSM (and also adipose tissue)?

6) In general, the authors should compare and state/discuss what tissue model is preferable for the development of O. volvulus and should be used for further research.

Minor comments:

1) Lines 196 and 275: O. volvulus in italics

2) Scale bar is missing in Figure 4D and E

3) Line 554: in vitro in italics

Reviewer #2: Comments to the authors

This manuscript reports on the development of an in-vitro 3-D tissue model to achieve in long-term cultures finally mature adult Onchocerca filariae for drug development and biomedical research. The authors aimed to provide a valuable source also of worms at various developmental stages. This investigation represents a highly relevant and promising innovative goal, but also a challenging ambition. The submitted data represent encouraging first results which have to be extended and further developed. The study and the sequence of experiments are largely scientifically sound, the results mostly clearly presented, figured in illustrations and photographs. Overall, this manuscript is relevant for future biomedical research on filariae and their control by novel drugs and should be published after revision to address all specified comments and concerns. 

Notably, the ambitious claims are highly demanding but should be to a degree toned. Thus, the title states „Optimizing 3-dimensional human skin models ...“ that interested helminthologists will assume that 3-D skin models have been already applied as source of developed nematode stages and in this report are further optimized. The title could read „First development of a 3-dimensional human skin models to facilitate long-term development of Onchocerca volvulus, a debilitating obligatory human parasite“. The authors claim to (i) establish a source of adult Onchocerca filariae, (ii) to contribute to the progress of anti-parasitic treatment strategies for various helminth infections, (iii) to support the development of new interventional tools to fight this debilitating obligatory human parasite - represent well-aimed future goals, (iv) to reveal a proof of concept - but the presented data constitute here first but very relevant and successful results for this aim. The intended long-term 3-D FTSM culture over multiple weeks at least appears very demanding and complex and further has to be elaborated and optimized.

The experiments widely comprise and compare 2-D culture conditions with 3-D tissue models exploring numerous media and larvae. The denotation, however, of the multiple applied media and their allocation to the numerous 2-D/3-D culture conditions appear considerably complex and rather transparent and should be improved. Further, the multitude of abbreviations has to be explained in a list..

Major points

The formation of an onchocercoma in the subcutaneous tissue of an infected host represents a host reaction to encapsulate the worm parasite and to prevent its further migration and development but foster its degeneration (PMID: 14738900, PMID: 1401100). Thus, the migration of adipocytes, mesenchymal stromal cell and dermal fibroblasts to the worm larvae and adherence of such host cells to the foreign surface are in agreement with the formation of the nodule. However, the accentuation that co-culturing of worm stages benefits to larval motility and fitness by assumed secreted factors as a positive result of the model - rather may reflect the host activity to constrain and cope with the invading parasite. Further, the complex full-thick skin model comprising epidermal keratinocytes, fibroblasts, stromal cells and adiposic cell types can admit the survival and development of L4 to young and subsequent mature adults – but this observation does not match the biological role of nodule formation which are one general goal of these experiments. In the infected host the worm stages can develop within in emerging onchocercoma rather by factors secreted by the parasite which hinder its restriction and damage – than by compounds secreted by host cells to foster and promote the parasites fitness and development, as suggested. The authors should reconsider the acting biological principles. There exists no biological regularity of the host to foster the fitness and development of its parasite. The authors should not mix and permute the natural biological processes and the target of these experiments namely to receive Onchocerca stages for (i) analysis of the parasite general biology (morphology, transcriptome and proteome profiling to validate their stage of development) as well as (ii) for immunological and vaccine experiments and (iii) for treatment trials.

The authors stated no detrimental effects on larvae by the applied stromal cells in the proximity of adipose aggregates, they have, however, to consider that in vivo onchocercoma innate and adaptive immune cells of the host can damage the worm stages by secreted toxic proteins and radicals (PMID: 2014134, PMID: 1980799, PMID: 6879705).

Critical is the formulation „L5, early pre-adult stage“ since according to the filariasis scientific community the developmental stages of all filariae are microfilaria – L1 – L2 – L3 – L4 – juvenile adult – mature adult. There are rather no significant publications on L5-early pre-adult stages but Voronin, 2019 (PMID: 30653499) and formerly Duke, 1991 (PMID: 1888206). A myriad of scientists of classical parasitology on filariasis had studied predominantly 20-50 years ago the developmental stages of Onchocerca as of other filariae (PMID: 9754295, PMC305371, PMID: 2794455, PMID: 20976099, PMID: 27881553, ...). The report from Duke (PMID: 1888206) could not be verified. In the recent publication molecular, immunological or cellular parasitologists (PMID: 30653499) did not reference the existence of pre-adult fifth-stage larvae. Detached cuticles have to be doubtless verified from both, L3 and subsequently from the generated L4. Of interest, the author Sarah Lustigman addressed L4 and next adult worms in a previous publication (PMID: 1582477).

Minor comments

As vitality and fitness criteria also the MTT assay can be applied based on the conversion of water soluble MTT (3-(4,5-dimethylthiazol-2-yl)-2,5-diphenyltetrazolium bromide) to an insoluble formazan product (PMID: 32628671).

Reference 2 should be appropriately PMID: 31723729.

Line 275: O. volvulus in italics

Reviewer #3: This manuscript reports progress on attempts to improve culture conditions for the development of Onchocerca volvulus larvae. There is a need for macrofilaricides (adulticides) and alternative approaches to ensure the eventual elminiation of river blindness. Adult parasites are difficult to source and culture, hampering efforts to directly study and screen this important life stage. No suitable small animal models support complete intra-mammalian development. This area of work is therefore highly significant, as it seeks to generate viable adult parasites from more readily sourced infective-stage larvae by optimizing in vitro culture approaches.

Previous work by the authors reported a 2D culture system that supported L4 to pre-adult (L5) development using feeder endothelial cells (HUVEC). Here, the authors explore the use of a hydrogel as a supporting matrix for 3D culture, co-culture with cell lineages matched to parasite niche, and direct contact between parasites and mammalian cells. Studies comparing four media conditions suitable for skin models, prioritized two media formulations (3D-3 and 3D-4) that led to better larval growth and motility. O. volvulus larvae were cultured with direct exposure to ECM and cells in a collagen-based skin model using a 'stamp' (mold) that allowed for some tracking of larvae. Indirect co-culturing was pursued to allow for better tracking, and also to allow the flow of cells and their secretions.

These experiments using the skin model are then supplanted by the adipose tissue model. Direct co-culture with adipose cells led to much more motile worms over a long time-course. The major limitations in this study are that the original goal of culturing viable adult stage worms has not been achieved, the primary molecular determinants of the improvements made are unclear, and the broad adoption of this protocol is made difficult by the requirement of primary cells. That said, these limitations are thoroughly discussed and this work provides a great deal of new data that moves this field forward. This work may also guide approaches to culture other filarial or skin-associated nematode parasites.

PLOS authors have the option to publish the peer review history of their article (what does this mean?). If published, this will include your full peer review and any attached files.

Reviewer #1: No

Reviewer #2: Yes: not sign

Reviewer #3: No
---

## [Decision Letter · Decision Letter 1]

4 Oct 2020

Dear Dr. Lustigman,

We are pleased to inform you that your manuscript 'Preliminary evaluations of 3-dimensional human skin models for their ability to facilitate in vitro the long-term development of the debilitating obligatory human parasite Onchocerca volvulus' has been provisionally accepted for publication in PLOS Neglected Tropical Diseases.

Best regards,

Keke Fairfax, PhD

Deputy Editor

Keke Fairfax

Deputy Editor

Reviewer's Responses to Questions

**Key Review Criteria Required for Acceptance?**

**Methods**

-Are the objectives of the study clearly articulated with a clear testable hypothesis stated?

-Is the study design appropriate to address the stated objectives?

-Is the population clearly described and appropriate for the hypothesis being tested?

-Is the sample size sufficient to ensure adequate power to address the hypothesis being tested?

-Were correct statistical analysis used to support conclusions?

-Are there concerns about ethical or regulatory requirements being met?

Reviewer #1: (No Response)

Reviewer #3: (No Response)

**Results**

-Does the analysis presented match the analysis plan?

-Are the results clearly and completely presented?

-Are the figures (Tables, Images) of sufficient quality for clarity?

Reviewer #1: (No Response)

Reviewer #3: (No Response)

**Conclusions**

-Are the conclusions supported by the data presented?

-Are the limitations of analysis clearly described?

-Do the authors discuss how these data can be helpful to advance our understanding of the topic under study?

-Is public health relevance addressed?

Reviewer #1: (No Response)

Reviewer #3: (No Response)

**Editorial and Data Presentation Modifications?**

Reviewer #1: (No Response)

Reviewer #3: (No Response)

**Summary and General Comments**

Reviewer #1: The authors responded to all raised questions from the reviewers sufficiently. I congratulate the authors to this work, which is highly relevant for the filarial research community.

Reviewer #3: The reviewers have responded constructively and addressed my concerns. I appreciate nuanced acknowledgements of the limitations of this study, while also understanding that parasite tissues and sourcing are a bottleneck to rapid improvements of this approach. All in all, this is a worthwhile study that will provide ideas to build upon, including by other groups working to improve culture conditions for other mammalian-parasitic nematodes.

PLOS authors have the option to publish the peer review history of their article (what does this mean?). If published, this will include your full peer review and any attached files.

Reviewer #1: No

Reviewer #3: No

---

## [Editor Report · Acceptance letter]

22 Oct 2020

Dear Dr. Lustigman,

We are delighted to inform you that your manuscript, "Preliminary evaluations of 3-dimensional human skin models for their ability to facilitate *in vitro* the long-term development of the debilitating obligatory human parasite *Onchocerca volvulus*," has been formally accepted for publication in PLOS Neglected Tropical Diseases.

Best regards,

Shaden Kamhawi

co-Editor-in-Chief

Paul Brindley

co-Editor-in-Chief
